# Consumers' Perception on Traceability of Greek Traditional Foods in the Post-COVID-19 Era

**Dimitris Skalkos** [1,*] **, Ioanna S. Kosma** [1] **, Eleni Chasioti** [1] **, Thomas Bintsis** [2] **and Haralabos C. Karantonis** [3]

1    Laboratory of Food Chemistry, Department of Chemistry, University of Ioannina, 45110 Ioannina, Greece; i.kosma@uoi.gr (I.S.K.); el.chasioti@uoi.gr (E.C.)
2    School of Science and Technology, Hellenic Open University, 11 Aristotelous, 54624 Thessaloniki, Greece; bintsis.thomas@ac.eap.gr
3    Laboratory of Food Chemistry, Biochemistry and Technology, Department of Food Science and Nutrition, School of the Environment, University of The Aegean, Metropolitan Ioakeim 2, 81400 Mytilene, Greece; chkarantonis@aegean.gr
*    Correspondence: dskalkos@uoi.gr; Tel.: +30-2651-008345

**Abstract:** In the rising new global economic and social period, after the COVID-19 pandemic, traceability is expected to be a critical parameter for the selection of foods by consumers worldwide. Accordingly, traditional foods (TFs) can become the foods of choice in the new era due to their originality, authenticity, unique organoleptic properties, and locality. In this paper, the consumers' perception on traceability regarding Greek TFs and northwest Greek TFs is investigated, in order to find out the specific information they require for the purchase of these foods. Traceability was tested using variables related to package, product, quality, process, and personal information of these foods. A self-response questionnaire survey was carried out in September and October 2021 on a sample of 1707 participants through the Google platform. The results show that the participants consider traceability regarding questions on package information "quite important" and "very important" by an average of 68%, on food information by 64%, on quality information by 69%, on production process information by 78%, and on personal information by 65%. A similar pattern was recorded for the regional northwest Greek TFs for information on production process, personal, and package data, although there was a significant increase in the perception by the participants for data related to food information itself by 87% and more related to quality information by 94%.

**Keywords:** traditional foods; traceability; package information; product information; quality information; process information; personal information; questionnaire survey; post-COVID-19 era

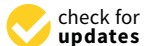

## 1. Introduction

Reports show that the economic crisis caused by the COVID-19 pandemic has a major impact on the global economy and significant changes will occur in the long run [1]. It affects all aspects of human life including the consumption of goods. There are signs of a growing anticonsumer movement, distinguished by Philip Kottler, of at least five types of anticonsumers [2]: the degrowth activists who feel that too much time and effort are going into consuming; the life simplifiers who want to eat less and buy less; the climate activists who worry about the damage to the planet through consumption; the food chooser who have turned into vegetarians and vegans; and the conservation activists who plead not to destroy existing goods but to reuse, repair, and redecorate them. These changes in the global dietary patterns introduce changes in the food production and supply processes as well. The highly globalized nature of today's food production and the supply commodities need to move from the world's source of grain supply to where they are consumed [3]. Internet and communication technologies, blockchain in the food supply chain and other industry 4.0 applications, as well as approaches that redefine the way we consume food, are the innovation with the highest potential in the new era [4]. There is also an equally

pressing need to exploit social marketing to understand attitudes, perceptions, and barriers that influence the behavior change of consumers and the agrifood industry [4]. Greece is a country which has experienced a sovereign debt crisis, similar to the coming global crisis, the results of which are currently under study at different levels such as the SMEs [5] or employees performance [6]. Subsequently, these changes will contribute to adapting to the new norms forged by the COVID-19 pandemic, where there is a significant gap in knowledge for decision making.

*Literature Review*

In the post-COVID-19 era, a major issue for the customers will be the traceability in the foods they choose to buy [7]. The history of a food product is the definition of food traceability, and it is important because it ensures valuable data for consumers [8]. Consumers' demand for information about the traceability of food products has increased significantly in the last decade due to market globalization and issues related to food quality, safety, trust, and environmental protection [9–13]. Credible information, good reputation, and, at the same time, the enhancement of consumers' welfare are interrelated aspects of brand performance included in the traceability frame [14]. These concepts are strongly related in consumers' minds, and, therefore, cannot be easily separated in explaining choices [15]. Food traceability can reduce information asymmetry and food safety risks [16–18]. Traceability depends on parameters connected to supply chain and to trade related issues [19–21]. The distrust to the governments worldwide is what makes traceability a valuable tool for increased consumer confidence in food safety [20,22]. A recent study proved that it is possible to affirm that disease/pest and inputs traceability are the elements that increase consumers' trust in food safety [23]. The results of another recent study in six China cities, just before the pandemic, showed that consumers are willing to pay for traceable food with strong evidence of preference heterogeneity and with their valuations differing upon the degree of their trust in government's supervision of food safety and food labels [24]. Another recent report studying the consumers' perspective on food origin traceability in Poland proved that parameters such as food product features, food product packaging information, and shopping place frequency are significant on tracing the food origin [25]. To prioritize drivers to create traceability in the food supply chain after COVID-19 pandemic, 16 drivers were identified and test-grouped into four groups of drivers as informational, environmental, social, and economic [7].

In the new post-COVID-19 era, traditional foods (TFs) can play a vital role as the food of choice for the anticonsumers described above, due to their particular characteristics and properties [26]. They have played an important role in the development of different cultures and regions [27]. Recent study proves that TFs in Europe have a role in food consumption [28]. They reflect cultural inheritance and have left their imprints on the contemporary dietary patterns [29]. The definition of the term "traditional" related to foods is provided by the European Union as "Tradition means proven usage in the community market for a time period showing transmission between generations; at least 25 years" [30]. TFs interfere between the consumers and producers, promoting cultural associations within each area [31]. Sensory attributes, gastronomic heritages, eating habits, and association with certain local areas are more characteristics of TFs [32–35]. The European Union has labeled TFs in three mini categories: PDO, protected designation of origin; PGI, protected geographical indication; and TGI, traditional specialty guaranteed [36]. EU regulation 1151/12 assists producers of TFs to communicate the products' characteristics and farming attributes to buyers and consumers [37]. The definition of the term "traditional" in the above document means proven usage on the domestic market for a period that allows transmission between generations, with the period being at least 30 years.

Greece uses the provisions of the Regulation in the national Legislation with Min-
isterial Decree (3321/145849) issued by the Hellenic Ministry of Food and Agricultural
Development since 2006 [38]. Registered traditional Greek foods by the different types are
shown in Table 1. Food and agriculture, producing mainly Greek TFs, make up 3.5% of
Greece GDP, the majority of which exported for consumption overseas in Europe, Russia,
the US, and elsewhere [39]. This brings an important revenue stream for the Greek economy
and keeps many farmers and food producers afloat.

**Table 1.** Distribution of Greek recognized foods between the different categories.

| Type of Food Products | PDOs | PGIs | TSGs |
|---|---|---|---|
| Wine | 33 | 116 | |
| Olive oil | 21 | | |
| Meat | 2 | | |
| Cheese | 22 | 1 | |
| Foods of animal origin | 2 | | |
| Fish | 1 | | |
| Fruits and vegetables | 29 | 21 | 1 |
| Others | 6 | | |
| Total | 116 | 138 | 1 |

We studied the consumers' trust in Greek TFs in the post-COVID-19 era and found
that they trust them since they "strongly agree" by an average of 20% and "agree" by an
average of 50% that TFs are safe, healthy, sustainable, authentic, and tasty [40].

TFs of northwest Greece (namely the region of Epirus) comprise a significant portion
of the overall Greek TFs. It is a region with local traditional food products as described
elsewhere [41] Selected regional TFs, mainly the PDO cheeses and wines, are exported
throughout Europe, thus promoting the regional brand name. Our recent results studying
the northwest Greece TFs indicate that the COVID-19 crisis has not interfered in consumers'
attitudes and perceptions regarding TFs [41].

We have shown the importance of the traceability of foods in the post-COVID-19
period and the potential of the TFs as the food of choice for the consumers of this new
period, namely the anticonsumers. The aim of the present work was to assess the five
determinants associated with the consumers' perception on the traceability of Greek TFs
in order to identify the key pieces of information required to ensure their future prospect,
growth, and development. These five determinants according to the existing literature
on food traceability [42–46] are information related to: package, product, quality, process,
and personal data. The current study examines these five determinants of consumers'
perception on the traceability of Greek TFs and the northwest Greek TFs in the post-
COVID-19 period:

(I)   Consumers' perception on **package information** of Greek TFs. This involves data
      regarding characteristics such as nutritional composition and energy value, expiration
      date, production date, and additional information required for the particular food;
(II)  Consumers' perception on **product information** of Greek TFs. This involves data
      regarding the origin, the producer, the brand name and the price of the food;
(III) Consumers' perception on **quality information** of Greek TFs. This involves data
      regarding quality label, certification label, safety label, and European origin label;
(IV)  Consumers' perception on **process information** of Greek TFs. This involves data
      regarding the method of production, the level of processing, the raw materials, and
      the additional ingredients used for the production of food;
(V)   Consumers' perception on **personal information** of Greek TFs. This involves data
      regarding pre-existing knowledge, recommendation by others, pre-existing personal
      experience, and origin of purchase.

In addition, the five determinants were examined on:

(VI) Consumers' perception on the **traceability of the Northwest Greek TFs** (Epirus' region).

This involves data regarding the package, product, quality, process, and personal data of the food mentioned in I–IV above.

## 2. Materials and Methods

### 2.1. Data Collection and Sample Characterization

A questionnaire to investigate the access data related to consumers' perception concerning traceability of Greek TFs, including TFs of Epirus region. The questionnaire included seven parts. The parts were built up using a similar previous study [46]. The first part included questions about the sociodemographic characteristics of the respondents. The second part consisted of four questions designed to assess the perception on the package information of the TFs, namely the nutritional and energy value, the production and expiration date, and the additional information which the participants would like to find in order to purchase them in the post COVID era. The third part included four questions focused on the participants' perception on the product information of the TFs, namely the producer, the geographic origin, the name, and the price of the TFs, which motivates their purchase. In the fourth part, issues concerning the participants' perception on the quality information of TFs, such as quality, certification, safety, and EU labels were assessed through four questions. The fifth part included four questions that approached the preferred process information data, namely the method of production, the level of processing, the raw material, and the ingredients used, of the participants in relation to their preference of TFs. In the sixth part, using four questions, the participants' preference on personal information of the TFs, such as pre-existing knowledge and personal experience, recommendation, and origin of purchase, regarding their perception was assessed. Finally, in the seventh part, using five questions, participants were asked to express their preference on traceability of the northwest Greece TFs, which can direct their purchasing choices. Issues such as package information, product information, quality information, process information, and personal information were taken into consideration. Quality of the data was obtained through the application of the questionnaire to 50 respondents who answered the questions easily. Electronic questionnaire was used. The distribution method chosen was by e-mail following the literature practices [47–49]. A snowball method was used to obtain a large number of participants [50]. The sample of the population is very well distributed among the different demographic characteristics, with participants familiar with the new technologies.

A higher rate for female respondents recorded at 61.7% is similar to the observation by other papers as well [51–54], leading to the conclusion that women respond more willingly to food-related surveys as they are primarily involved in the household organization. The research questionnaire was created through the Google platform and the Google Forms function. The geographical context for the present study was all the Greek regions, divided into five parts: north, west, central, south, and the islands, since the country includes many of them in the Aegean and the Ionian seas. The sample included students, among others. The participants received information explaining the purpose of the research, while obtaining access to the electronic form of the questionnaire through an attached link.

The survey took place during the period September–October 2021 and consisted of 1707 participants (Table 2).

**Table 2.** Sociodemographic characterization of the sample.

| Variable | Groups | (%) |
|---|---|---|
| *Gender* | Male | 38.3 |
| | Female | 61.7 |
| *Age* | 18–25 | 38.9 |
| | 26–35 | 10.0 |
| | 36–45 | 12.9 |
| | 46–55 | 22.2 |
| | 56+ | 16.0 |
| *Level of education* | None/primary school | 0.2 |
| | Secondary school | 0.4 |
| | High school | 7.1 |
| | University | 92.3 |
| *Civil state* | Single | 53.7 |
| | Married | 42.0 |
| | Divorced | 3.7 |
| | Widow/widower | 0.5 |
| *Job situation* | Employed | 55.7 |
| | Unemployed | 3.5 |
| | Student | 37.6 |
| | Retired | 3.1 |
| *Permanent resident in Greece* | NORTH GREECE (regions of Macedonia—Thrace) | 27.7 |
| | WEST GREECE (region of Epirus—Etoloakarnania prefecture) | 25.2 |
| | CENTRAL GREECE (including Athens) | 35.4 |
| | SOUTH GREECE (region of Peloponnese) | 3.7 |
| | ISLANDS (Ionian and Aegean) | 8.0 |

From the 1707 participants, 38.3% were male and 61.7% female. Regarding the spatial distribution, 25.2% were permanent residents of west Greece, 35.4% of central Greece (including the capital Athens), 27.7% residents of north Greece, 8.0% residents of the Greek islands, and 3.7% of south Greece, leading to a wide geographic distribution. The majority of the participants were aged between 18–25, 46–55, and 56+ years (38.9%, 22.2%, and 16.0%, respectively), while the other age groups, 26–35 and 36–45, were the least represented (10.0% and 12.9%, respectively). Regarding the level of education, most of the participants had higher education (university, 92.3%), and only 0.6% had completed primary or secondary school, while the employment status category was dominated by employed (55.7%), and students (37.8%) participants. Regarding the civil state of the participants, 42.0% were married, 53.7% were single, 3.7% were divorced, and only 0.5% were widows. It is worth mentioning that there was a significant percentage of young participants (students, at the age of 18–25) in the study which gives a better prospective, value to the results obtained, since the new generation better shows the trends of the future.

### 2.2. Data Analysis

Basic statistical tools were used for the exploratory analysis of the data. The survey was prepared in Greek and divided into seven parts, as detailed above:

Part I.    Sociodemographic data;
Part II.   Consumers' perception on the package information of Greek TFs;
Part III.  Consumers' perception on the product information of Greek TFs;
Part IV.   Consumers' perception on the quality information of Greek TFs;
Part V.    Consumers' perception on the process information of Greek TFs;
Part VI.   Consumers' perception on the personal information of Greek TFs;
Part VII.  Consumers' perception on the traceability of northwest Greek TFs.

In order to measure the respondents' opinion about a set of statements related to TFs, a 5-point Likert scale, ranging from 1 = not at all important, 2 = less important, 3 = moderately important, 4 = quite important, and 5 = very important to me, was used [55].

Details of the statistics performed have recently been described in detail [40]. The Cramer's V coefficient used, ranging from 0 to 1, can be interpreted as follows: $V \approx 0.1$

weak association, $V \approx 0.3$ moderate association, and $V \approx 0.5$ or over, strong association. In all the tests performed, the level of significance considered was 5% ($p < 0.05$).

## 3. Results

In the results presented in the tables below, the percentages of not at all important (1) and less important (2) are less than 15% and considered minor, and no specific attention is given to all of them. Table 3 presents the participants' perception on package information of Greek TFs. The results show that the majority of the participants find the information about the expiration date (76.1%) and nutritional value (52.2%) to be very important, while a significant portion finds the information concerning the date of production (38.6%) to be very important, while they are not interested in additional information provided (12.9%).

**Table 3.** Participants' perception on **package information** of Greek TFs (scale from 1 = not at all important to 5 = very important).

| Questions<br>How Important Is the Information on the Food Package to You, Regarding | Answers According to Scale Points (%) | | | | |
|---|---|---|---|---|---|
| | 1 | 2 | 3 | 4 | 5 |
| 1. The nutritional composition and energy value | 1.6 | 3.4 | 14.3 | 28.6 | 52.2 |
| 2. The best before date | 0.5 | 1.7 | 5.6 | 16.2 | 76.1 |
| 3. The date of production | 4.4 | 9.3 | 20.7 | 27.0 | 38.6 |
| 4. The access to additional information (by the use of a phone number or website) | 16.4 | 19.4 | 29.3 | 22.0 | 12.9 |

The chi-square test presented in Table 4 shows that there were significant differences between consumers' perceptions on package information of Greek TFs in terms of:

1. Nutritional value of Greek TFs: between age ($x^2 = 78.366$, $p = 0.000$), level of education ($x^2 = 57.565$, $p = 0.000$), civil state ($x^2 = 60.294$, $p = 0.000$), and job situation ($x^2 = 66.550$, $p = 0.000$).
2. Best before date of Greek TFs: between gender ($x^2 = 24.264$, $p = 0.007$), level of education ($x^2 = 62.914$, $p = 0.000$), and civil state ($x^2 = 21.092$, $p = 0.049$).
3. Date of production of Greek TFs: between gender ($x^2 = 10.268$, $p = 0.036$), age ($x^2 = 121.564$, $p = 0.000$), level of education ($x^2 = 25.896$, $p = 0.011$), civil state ($x^2 = 94.153$, $p = 0.000$), and job situation ($x^2 = 93.529$, $p = 0.001$).
4. Access to additional information: between age ($x^2 = 11.918$, $p = 0.018$), age ($x^2 = 118.338$, $p = 0.000$), level of education ($x^2 = 22.240$, $p = 0.035$), civil state ($x^2 = 72.014$, $p = 0.000$), and job situation ($x^2 = 90.281$, $p = 0.002$).

Table 5 presents the participants' perception on product information of Greek TFs. The results show that 55.3% of the participants find the price of the product to be very important and 33.5% find its geographical origin to be very important. The name of the product and the identification of the producer are of less importance (by 21.6% and 20.3% for the very important answer, respectively).

The chi-square test presented in Table 4 showed that there were significant differences between consumers' perceptions on product information of Greek TFs in terms of:

1. Identification of the producer: between age ($x^2 = 83.269$, $p = 0.000$), civil state ($x^2 = 51.557$, $p = 0.000$), and job situation ($x^2 = 72.686$, $p = 0.000$).
2. Geographic origin of the food: between gender ($x^2 = 13.580$, $p = 0.009$), age ($x^2 = 145.520$, $p = 0.000$), civil state ($x^2 = 88.211$, $p = 0.000$), and job situation ($x^2 = 105.982$, $p = 0.000$).
3. Name of the product (branding): between age ($x^2 = 34.549$, $p = 0.005$), civil state ($x^2 = 28.105$, $p = 0.005$), job situation ($x^2 = 25.226$, $p = 0.014$), and residency ($x^2 = 27.619$, $p = 0.035$).
4. Price of the product: between age ($x^2 = 20.552$, $p = 0.000$), age ($x^2 = 64.024$, $p = 0.000$), level of education ($x^2 = 69.607$, $p = 0.000$), job situation ($x^2 = 44.006$, $p = 0.000$), and residency ($x^2 = 29.137$, $p = 0.023$).

**Table 4.** Association between variables (A) package, (B) product, and (C) quality information of Greek TFs and the sociodemographic variables.

| | Gender | | | Age | | | Level of Education | | | Civil State | | | Job Situation | | | Residency | | |
|---|---|---|---|---|---|---|---|---|---|---|---|---|---|---|---|---|---|---|
| | $X^2$ * | *p* ** | V *** | $X^2$ | *p* | V | $X^2$ | *p* | V | $X^2$ | *p* | V | $X^2$ | *p* | V | $X^2$ | *p* | V |
| **A. Package information of Greek TFs** | | | | | | | | | | | | | | | | | | |
| 1. The nutritional and energy value | | | | 78.366 | 0.000 | 0.215 | 57.565 | 0.000 | 0.184 | 60.294 | 0.000 | 0.190 | 66.550 | 0.000 | 0.199 | | | |
| 2. The best before date | 24.264 | 0.000 | 0.120 | | | | 62.914 | 0.000 | 0.193 | 21.092 | 0.049 | 0.112 | | | | | | |
| 3. The date of production | 10.268 | 0.036 | 0.078 | 121.564 | 0.000 | 0.268 | 25.896 | 0.011 | 0.124 | 94.153 | 0.000 | 0.237 | 93.529 | 0.001 | 0.236 | | | |
| 4. The access to additional information | 11.918 | 0.018 | 0.084 | 118.338 | 0.000 | 0.264 | 22.240 | 0.035 | 0.115 | 72.014 | 0.000 | 0.207 | 90.281 | 0.000 | 0.231 | | | |
| **B. Product information of Greek TFs** | | | | | | | | | | | | | | | | | | |
| 1. The identification of the producer | | | | 83.269 | 0.000 | 0.221 | | | | 51.557 | 0.000 | 0.175 | 72.686 | 0.000 | 0.207 | | | |
| 2. The geographic origin of the food | 13.580 | 0.009 | 0.089 | 145.520 | 0.000 | 0.293 | | | | 88.211 | 0.000 | 0.229 | 105.982 | 0.000 | 0.250 | | | |
| 3. The name of the product (branding) | | | | 34.549 | 0.005 | 0.143 | | | | 28.105 | 0.005 | 0.130 | 25.226 | 0.014 | 0.122 | 27.619 | 0.035 | 0.128 |
| 4. The price of the food | 20.552 | 0.000 | 0.110 | 64.024 | 0.000 | 0.194 | 69.607 | 0.000 | 0.203 | | | | 44.006 | 0.000 | 0.162 | 29.137 | 0.023 | 0.132 |
| **C. Quality information of Greek TFs** | | | | | | | | | | | | | | | | | | |
| 1. The quality label of the product | | | | 40.928 | 0.001 | 0.155 | | | | 33.248 | 0.001 | 0.141 | 26.728 | 0.008 | 0.126 | | | |
| 2. The certification label/logo | 10.759 | 0.029 | 0.080 | 79.525 | 0.000 | 0.217 | | | | 62.227 | 0.000 | 0.193 | 59.218 | 0.000 | 0.188 | | | |
| 3. The safety label | 14.472 | 0.006 | 0.093 | 102.316 | 0.000 | 0.246 | 25.035 | 0.015 | 0.122 | 77.325 | 0.000 | 0.215 | 95.851 | 0.000 | 0.239 | 29.538 | 0.021 | 0.133 |
| 4. The European origin label | 14.388 | 0.005 | 0.092 | 172.109 | 0.000 | 0.319 | | | | 128.583 | 0.000 | 0.277 | 125.554 | 0.000 | 0.273 | | | |

* chi-square test, ** level of significance of 5%: $p < 0.05$, *** Cramer's coefficient.

**Table 5.** Participants' perception on **product information** of Greek TFs (Scale from 1 = not at all important to 5 = very important).

| Questions | Answers According to Scale Points (%) | | | | |
|---|---|---|---|---|---|
| How Important Is the Information of the Food to You, Regarding | 1 | 2 | 3 | 4 | 5 |
| 1. The identification of the producer | 5.9 | 11.2 | 33.0 | 29.5 | 20.3 |
| 2. The geographic origin of the food | 3.5 | 8.5 | 20.8 | 33.7 | 33.5 |
| 3. The name of the product (branding) | 5.3 | 10.5 | 30.8 | 31.9 | 21.6 |
| 4. The price of the food | 0.5 | 2.1 | 12.2 | 29.9 | 55.3 |

Table 6 presents the participants' perception on quality information of Greek TFs. The results show that consumers find all of the above information to be quite important and very important. Specifically, the safety label as well as the certification logo seem to be a very important information that concerns them by 39.4% and by 37.4%, respectively, followed closely by the quality label and the European origin label (35.2% and 36.7%, respectively, with very important information as an answer).

**Table 6.** Participants' perception on **quality information** of Greek TFs (Scale from 1 = not at all important to 5 = very important).

| Questions | Answers According to Scale Points (%) | | | | |
|---|---|---|---|---|---|
| How Important Is the Information of the Food Quality to You, Regarding | 1 | 2 | 3 | 4 | 5 |
| 1. The quality label of the product (i.e., retailer quality label, national quality label, quality label of organizations, etc.) | 1.9 | 5.9 | 20.0 | 37.0 | 35.2 |
| 2. The certification label/logo (i.e., ECO label, etc.) | 4.3 | 8.1 | 18.9 | 31.3 | 37.4 |
| 3. The safety label (i.e., salmonella free, ISO, safety checked, etc.) | 5.6 | 9.0 | 19.0 | 27.0 | 39.4 |
| 4. The European origin label (PDO, PGI, and TSG) | 5.3 | 7.9 | 19.1 | 31.0 | 36.7 |

The chi-square test presented in Table 4 shows that there were significant differences between consumers' perceptions on quality information of Greek TFs in terms of:

1. Quality label of the product: between age ($x^2$ = 40.928, $p$ = 0.001), civil state ($x^2$ = 33.248, $p$ = 0.001), and job situation ($x^2$ = 26.728, $p$ = 0.008).
2. Certification label/logo: between gender ($x^2$ = 10.759, $p$ = 0.029), age ($x^2$ = 79.525, $p$ = 0.000), civil state ($x^2$ = 62.227, $p$ = 0.000), and job situation ($x^2$ = 59.218, $p$ = 0.000).
3. Safety label: between gender ($x^2$ = 14.472, $p$ = 0.006), age ($x^2$ = 102.316, $p$ = 0.000), level of education ($x^2$ = 25.035, $p$ = 0.015), civil state ($x^2$ = 77.325, $p$ = 0.000), job situation ($x^2$ = 95.851, $p$ = 0.000), and residency ($x^2$ = 29.538, $p$ = 0.021).
4. European origin label: between gender ($x^2$ = 14.388, $p$ = 0.005), age ($x^2$ = 172.109, $p$ = 0.000), civil state ($x^2$ = 128.583, $p$ = 0.000), and job situation ($x^2$ = 125.554, $p$ = 0.000).

Table 7 presents the participants' perception on process information of Greek TFs. The results show that more than 50% of the participants find the information to be very important about the raw materials used (56.8%) and the other ingredients used (58.8%), the additives (58.8%) used for the production process. On the other, participants seem to believe that the method of production and the level of processing is information of less importance (33.4%, and 35.6%, respectively, with very important information as the answer of choice).

The chi-square test presented in Table 8 showed that there were significant differences between consumers' perceptions on process information of Greek TFs in terms of:

1. The used method of production: between age ($x^2$ = 147.852, $p$ = 0.000), civil state ($x^2$ = 83.269, $p$ = 0.000), and job situation ($x^2$ = 103.922, $p$ = 0.000).
2. The level of processing: between age ($x^2$ = 61.676, $p$ = 0.000), civil state ($x^2$ = 36.345, $p$ = 0.000), and job situation ($x^2$ = 49.155, $p$ = 0.000).
3. The raw materials used: between gender ($x^2$ = 9.877, $p$ = 0.043), age ($x^2$ = 189.659, $p$ = 0.000), civil state ($x^2$ = 128.570, $p$ = 0.000), job situation ($x^2$ = 158.369, $p$ = 0.001), and residency ($x^2$ = 29.258, $p$ = 0.022).

4. The ingredients used: between age ($x^2$ = 166.051, $p$ = 0.000), civil state ($x^2$ = 100.133, $p$ = 0.000), and job situation ($x^2$ = 119.588, $p$ = 0.000).

**Table 7.** Participants' perception on **process information** of Greek TFs (Scale from 1 = not at all important to 5 = very important).

| Questions | Answers According to Scale Points (%) | | | | |
|---|---|---|---|---|---|
| How Important Is the Information about the Process of the Food to You, Regarding | 1 | 2 | 3 | 4 | 5 |
| 1. The used method of production (e.g., organic production, etc.) | 2.8 | 7.7 | 19.7 | 33.4 | 36.5 |
| 2. The level of processing (e.g., whole tomato or tomato soup, etc.) | 2.4 | 7.2 | 19.5 | 35.3 | 35.6 |
| 3. The raw materials the food is made from | 1.1 | 3.4 | 11.0 | 27.6 | 56.8 |
| 4. The ingredients used | 1.4 | 3.5 | 10.2 | 26.1 | 58.8 |

Table 9 presents the participants' perception on personal information of Greek TFs. The results show that none of these pieces of information are very important by a major percentage, i.e., more that 50% of the participants. They find by 83.7% the pre-existing experience concerning the TFs as quite and very important (41.7% quite important and 42% very important) and by 77% the pre-existing knowledge (43.2% quite important and 33.8% very important). On the other hand, consumers find the recommendation by others to be moderately important (35.8%) and quite important (37.8%). Finally, the origin of purchase seems to be moderately important for 29.0% and quite important for 30.6% of the participants.

The chi-square test presented in Table 8, showed that there were significant differences between consumers' perceptions on personal information of Greek TFs in terms of:

1. Pre-existing knowledge: between age ($x^2$ = 79.875, $p$ = 0.000), level of education ($x^2$ = 26.582, $p$ = 0.009), civil state ($x^2$ = 53.269, $p$ = 0.000), and job situation ($x^2$ = 59.155, $p$ = 0.000).
2. Recommendation by friends and family: only between gender ($x^2$ = 19.569, $p$ = 0.001.
3. Pre-existing personal experience: between gender ($x^2$ = 11.344, $p$ = 0.023), age ($x^2$ = 30.045, $p$ = 0.018), level of education ($x^2$ = 33.175, $p$ = 0.001), and job situation ($x^2$ = 23.021, $p$ = 0.028).
4. Origin of purchase (e.g., super market, minimarket, grocery store, and market place): only between gender ($x^2$ = 24.299, $p$ = 0.000).

Table 10 presents the participants' perception on traceability of northwest (the region of Epirus) Greek TFs. The results show that 71.8% of the participants find the information concerning the quality of the food to be very important, while the information of the food itself greatly concerns 55.1%. Package data on the other hand seems to be a moderately important information for 31.6% and quite important for 36.0%. Finally, production process information is very important for 39.5% and personal experience for 37.4% of the participants.

The chi-square test presented in Table 8 shows that there were significant differences between consumers' perceptions on traceability of northwest Greek TFs in terms of:

1. Package data: between civil state ($x^2$ = 22.322, $p$ = 0.034), job situation ($x^2$ = 22.391, $p$ = 0.033), and residency ($x^2$ = 28.887, $p$ = 0.025).
2. The food itself: between age ($x^2$ = 35.910, $p$ = 0.003), level of education ($x^2$ = 22.594, $p$ = 0.031), civil state ($x^2$ = 21.561, $p$ = 0.043), job situation ($x^2$ = 24.512, $p$ = 0.017), and residency ($x^2$ = 26.526, $p$ = 0.047).
3. Quality of the food: between gender ($x^2$ = 14.008, $p$ = 0.007), age ($x^2$ = 32.754, $p$ = 0.008), level of education ($x^2$ = 126.505, $p$ = 0.000), civil state ($x^2$ = 23.125, $p$ = 0.027), and job situation ($x^2$ = 22.782, $p$ = 0.030).
4. Production process: between age ($x^2$ = 119.974, $p$ = 0.000), level of education ($x^2$ = 33.557, $p$ = 0.001), civil state ($x^2$ = 77.535, $p$ = 0.000), and job situation ($x^2$ = 91.718, $p$ = 0.000).
5. Personal experience: between gender ($x^2$ = 23.834, $p$ = 0.000), age ($x^2$ = 33.107, $p$ = 0.007), level of education ($x^2$ = 62.662, $p$ = 0.000), civil state ($x^2$ = 25.440, $p$ = 0.013), and job situation ($x^2$ = 31.843, $p$ = 0.001).

**Table 8.** Association between variables (A) process, (B) personal information of Greek TFs, (C) traceability of northwest Greek TFs, and the sociodemographic variables.

| | Gender | | | Age | | | Level of Education | | | Civil State | | | Job Situation | | | Residency | | |
|---|---|---|---|---|---|---|---|---|---|---|---|---|---|---|---|---|---|---|
| | $X^2$ * | $p$ ** | V *** | $X^2$ | $p$ | V | $X^2$ | $p$ | V | $X^2$ | $p$ | V | $X^2$ | $p$ | V | $X^2$ | $p$ | V |
| **A. Process information of Greek TFs** | | | | | | | | | | | | | | | | | | |
| 1. The used method of production | | | | 147.852 | 0.000 | 0.296 | | | | 83.269 | 0.000 | 0.223 | 103.922 | 0.000 | 0.248 | | | |
| 2. The level of processing | | | | 61.676 | 0.000 | 0.191 | | | | 36.345 | 0.000 | 0.147 | 49.155 | 0.000 | 0.171 | | | |
| 3. The raw materials the food is made from | 9.877 | 0.043 | 0.076 | 189.659 | 0.000 | 0.335 | | | | 128.570 | 0.000 | 0.277 | 158.369 | 0.001 | 0.306 | 29.258 | 0.022 | 0.132 |
| 4. The ingredients used | | | | 166.051 | 0.000 | 0.313 | | | | 100.133 | 0.000 | 0.244 | 119.588 | 0.000 | 0.266 | | | |
| **B. Personal information of Greek TFs** | | | | | | | | | | | | | | | | | | |
| 1. Pre-existing knowledge | | | | 79.875 | 0.000 | 0.217 | 26.582 | 0.009 | 0.125 | 53.269 | 0.000 | 0.178 | 59.155 | 0.000 | 0.187 | | | |
| 2. Recommendation by friends and family | 19.569 | 0.001 | 0.108 | | | | | | | | | | | | | | | |
| 3. Pre-existing personal experience | 11.344 | 0.023 | 0.082 | 30.045 | 0.018 | 0.133 | 33.175 | 0.001 | 0.140 | | | | 23.021 | 0.028 | 0.117 | | | |
| 4. Product origin of purchase | 24.299 | 0.000 | 0.120 | | | | | | | | | | | | | | | |
| **C. Traceability information of northwest Greek TFs** | | | | | | | | | | | | | | | | | | |
| 1. Package data | | | | | | | | | | 22.322 | 0.034 | 0.115 | 22.391 | 0.033 | 0.115 | 28.887 | 0.025 | 0.131 |
| 2. The food itself | | | | 35.910 | 0.003 | 0.146 | 22.594 | 0.031 | 0.116 | 21.561 | 0.043 | 0.113 | 24.512 | 0.017 | 0.121 | 26.526 | 0.047 | 0.126 |
| 3. Quality of the food | 14.008 | 0.007 | 0.091 | 32.754 | 0.008 | 0.139 | 126.505 | 0.000 | 0.274 | 23.125 | 0.027 | 0.118 | 22.782 | 0.030 | 0.116 | | | |
| 4. Production process | | | | 119.974 | 0.000 | 0.267 | 33.557 | 0.001 | 0.141 | 77.535 | 0.000 | 0.215 | 91.718 | 0.000 | 0.234 | | | |
| 5. Personal experience with the food | 23.834 | 0.000 | 0.119 | 33.107 | 0.007 | 0.140 | 62.662 | 0.000 | 0.193 | 25.440 | 0.013 | 0.123 | 31.843 | 0.001 | 0.137 | | | |

* chi-square test, ** level of significance of 5%: $p < 0.05$, *** Cramer's coefficient.

**Table 9.** Participants' perception on **personal information** of Greek TFs (scale from 1 = not at all important to 5 = very important).

| Questions<br>How Important Is the Personal Information of the Food to You, Regarding | Answers According to Scale Points (%) | | | | |
| --- | --- | --- | --- | --- | --- |
| | 1 | 2 | 3 | 4 | 5 |
| 1. Pre-existing knowledge | 0.9 | 3.1 | 18.9 | 43.2 | 33.8 |
| 2. Recommendation by friends and family | 3.5 | 12.2 | 35.8 | 37.8 | 10.7 |
| 3. Pre-existing personal experience | 0.5 | 1.5 | 14.2 | 41.7 | 42.0 |
| 4. Product origin of purchase (e.g., super market, mini market, grocery store, and market place) | 8.2 | 13.4 | 29.0 | 30.6 | 18.8 |

**Table 10.** Participants' perception on **traceability** of the northwest Greek TFs (Scale from 1 = not at all important to 5 = very important).

| Questions<br>Based on the above 5 Categories, How Important Is the Information of the Northwest Greek TFs to You, Regarding | Answers According to Scale Points (%) | | | | |
| --- | --- | --- | --- | --- | --- |
| | 1 | 2 | 3 | 4 | 5 |
| 1. Package data | 3.4 | 9.8 | 31.6 | 36.0 | 19.3 |
| 2. The food itself | 0.5 | 1.9 | 10.3 | 32.2 | 55.1 |
| 3. Quality of the food | 0.4 | 0.6 | 5.0 | 22.2 | 71.8 |
| 4. Production process | 2.0 | 4.9 | 18.6 | 35.0 | 39.5 |
| 5. Personal experience with the food | 0.8 | 2.6 | 18.9 | 40.3 | 37.4 |

## 4. Discussion

In this research, the consumer's perception regarding the five main traceability determinants of TFs, specifically of Greek TFs, after the COVID-19 pandemic is investigated for the first time (package/product/quality/process/personal information). Greek TFs have a long tradition of increased production and use, and it is for this reason that they were chosen for this study. In addition, the northwest Greek TFs, from the region of Epirus, were also chosen for comparison reasons, since this is a typical Greek mountainous, environmentally intact region with increased TFs and significant recognition by Greek consumers, as we have proved recently [41]. The sociodemographic characteristics of the participants of the survey exhibited in accordance to the literature [56]. They were from all different parts of Greece in order to ensure geographical distribution as well.

The package information data chosen in this study had a positive perception by the participants by more than 65%, except for the access to additional information which had only 34.9% (Table 3). The results of the chi-square test indicated that there were significant differences regarding package information between: (a) "gender" regarding the best before date, date of production, and access to additional information with weak association (V = 0.120/0.078/0.084); (b) "age" regarding nutritional value, date of production, and access to additional info with weak to moderate association (V = 0.215/0.268/0.264); (c) "level of education" regarding nutritional value, best before date, date of production, and access to additional info with weak association (V = 0.184/0.193/0.124/0.115); (d) "civil state" regarding nutritional value, best before date, date of production, and access to additional info with weak to moderate association (V = 0.190/0.112/0.237/0.207); and (e) "job situation" regarding nutritional value, date of production, and access to additional info with weak to moderate association (V = 0.199/0.236/0.231). Consumers' recognition and understanding on package information of food, especially regarding production, have been studied for more than a decade and proven to be important for their preference [57]. Nutritional knowledge are broadly helpful improving the accuracy of product choices, regardless of personal factors such as age, education, sex, etc. [58]. The legibility of food package information appears to be an equal challenge for young and elderly consumers [59]. Consumers value the best before date and production date as important information regarding their final decision to throw or not the food away [60]. These results agree with our finding regarding the importance of package information.

Overall, the participants consider the chosen product information to be important, although there is significant difference in their perception as shown in Table 4. The results of the chi-square test indicated that there were significant differences regarding product information between: (a) "gender" regarding the geographic origin of the food and its price with weak association (V = 0.089/0.110); (b) "age" regarding the identification of the producer, the geographic origin of the food, the name of the product, and its price with weak to moderate association (V = 0.221/0.293/0.143/0.194); (c) "level of education" regarding the price of the food with weak to moderate association (V = 0.203); (d) "civil state" regarding the identification of the producer, the geographic origin of the food, and the name of the product with weak to moderate association (V = 0.175/0.229/0.130); and (e) "job situation" regarding the identification of the producer, the geographic origin of the food, the name of the product, and its price with weak to moderate association (V = 0.207/0.250/0.122/0.162); and (f) "residency" regarding the name of the product and its price with weak association (V = 0.128/0.132). The importance of food price by the consumers has been studied extensively in the past [61–63]. A recent study proves that the price of food items is sometimes the only consideration when selecting food products, irrespective of their perceived quality and nutritional value [64]. The literature has examined consumers' preference for food of specific origin mainly country of origin and found a strong positive impact [65,66]. A recent report finds significant consumers' acceptance and preference for the Artic regional food products of Canada [67]. The name of the products (brands) especially for national and private brands of foods as reported this year have the same positive impact [68]. Our data prove the validity of the literature findings on Greek TFs as well.

The positive results regarding participants' perception on quality information of Greek TFs, shown in Table 5, are similar for all the issues addressed, in the range of 66–68%, as shown in Table 5. The results of the chi-square test indicated that there were significant differences regarding quality information between: (a) "gender" regarding the certification label, the safety label, and the European origin label with weak association (V = 0.080/0.093/0.092); (b) "age" regarding the quality label of the product, the certification label, the safety label, and the European origin label with weak to moderate association (V = 0.155/0.217/0.246/0.319); (c) "level of education" regarding the safety label with weak association (V = 0.122); (d) "civil state" regarding the quality label of the product, the certification label, the safety label, and the European origin label with weak to moderate association (V = 0.141/0.193/0.215/0.277); (e) "job situation" regarding the quality label of the product, the certification label, the safety label, and the European origin label with weak association (V = 0.126/0.188/0.239/0.273); and (f) "residency" regarding the safety label with weak association (V = 0.133). Recent reports indicate that different consumer segments have different attitudes and perceptions regarding food quality labels [69] and that PDO and organic labels are considered both labels substitutes by the majority of consumers [70]. These findings are also in agreement with our results regarding Greek TFs as well.

Increased results for traceability were recorded for the process information selected as shown in Table 7 above ranging from 70% to 85%. The results of the chi-square test indicated that there were significant differences regarding process information of the TFs between: (a) "gender" regarding the raw materials used with weak association (V = 0.076); (b) "age" regarding the used method of production, the level of processing, the raw materials, and the ingredients used with weak to moderate association (V = 0.296/0.191/0.335/0.313); (c) "civil state" regarding the used method of production, the level of processing, the raw materials, and the ingredients used with weak to moderate association (V = 0.233/0.147/0.277/0.244); (d) "job situation" regarding the used method of production, the level of processing, the raw materials, and the ingredients used with weak to moderate association (V = 0.248/0.171/0.306/0.266); and (a) "residency" regarding the raw materials used with weak association (V = 0.132).

When it comes to personal information regarding the traceability parameters for Greek TFs of choice, as shown in Table 8, positive results over 50% were recorded for

pre-existing knowledge and pre-existing personal experience, while results slightly less than 50% were recorded for the origin of purchase and recommendation by friend and family. The results of the chi-square test indicated that there were significant differences regarding personal information between: (a) "gender" regarding the recommendation by others, the pre-existing personal experience, and the origin of purchase with weak association (V = 0.108/0.082/0.120); (b) "age" regarding the pre-existing knowledge, and the pre-existing personal experience with weak association (V = 0.217/0.133); (c) "level of education" regarding the pre-existing knowledge and the pre-existing personal experience with weak association (V = 0.125/0.140); (d) "civil state" regarding the pre-existing knowledge with weak association (V = 0.178); and (e) "job situation" regarding the pre-existing knowledge and the pre-existing personal experience with weak association (V = 0.187/0.177). The reports in the literature on personal knowledge of food preference specify them on different items such as safety, hygiene, etc. [71–74], therefore cannot be compared with our TFs findings.

The participants' perception on the traceability parameters of the Epirus' Greek TFs, as shown in Table 9, compared with their perception for Greek TFs followed the similar pattern for most of the five parameters tested, except the quality information and the food information, as shown in Table 9. The results of the chi-square test indicated that there were significant differences regarding traceability information on northwest Greek TFs between: (a) "gender" regarding the quality of the food and personal experience with weak association (V = 0.091/0.119); (b) "age" regarding the food itself, quality of the food, the production process, and personal experience with weak association (V = 0.146/0.139/0.267/0.140); (c) "level of education" regarding the food itself, quality of the food, the production process, and personal experience with weak to moderate association (V = 0.116/0.274/0.141/0.193); (d) "civil state" regarding package data, the food itself, quality of the food, the production process, and personal experience with weak association (V = 0.115/0.113/0.118/0.215/0.123); (e) "job situation" regarding package data, the food itself, quality of the food, the production process, and personal experience with weak association (V = 0.115/0.121/0.116/0.234/0.137); and (f) "residency" regarding package data and the food with weak association (V = 0.131/0.126). Recent results suggest that COVID-19 psychological pressure was associated with an impulsive approach to buying food [75]. Consequently, it is of major importance to predict whether or not the food-purchasing behavior reverts to pre-COVID-19 habits when the emergency is over or it takes another path in the new rising economy.

## 5. Conclusions

This research work explores the consumers' perception on the five main determinants of traceability of Greek TFs at the beginning of the new post-COVID-19 era. The study applied these parameters on food traceability of the TFs in the Greek consumers' mind in order to find the parameters that are significant to their preference for information regarding the purchase of TFs. A questionnaire was completed by 1707 Greek participants conducted in September and October 2021. The present pandemic is causing major changes on consumers' mind and preferences, which is leading to changes of their selection of foods in an unprecedented way under investigation currently. With a relevant degree of uncertainty, it is believed that people will be more selected on food, especially the new generation of anticonsumers, purchasing it in a personalized way, with a focus on the environmental, health, and safety effects. Our results show that the participants of this study appreciate, in the order of importance, the information regarding production > process > quality > package data > personal > food itself available as traceability characteristics in order to consider them the food of choice in the future. Participants express their satisfaction with the package of these five characteristics associated with the TFs.

In order to evaluate the possible regional originalities and characteristics of the consumers' evaluation on the traceability of Greek TFs, a regional TF group of products, namely the northwest Greece (region of Epirus), TFs were used at the end of the same

survey with the same participants. The results showed that customers perceived in a similar manner the Epirus' TFs, information on production process, personal data, and package data, while the quality and the food itself data were considered more important as compared to the corresponding issues of the overall Greek TFs.

In the study, more women, educated, and employed participants, as well as young students, took place in the survey, and this can be considered a limitation of the study, even though the number of responses obtained is considered adequate. In addition, a limitation of the study is the use of Greek TFs only without the use of TFs by other countries which can have a different impact to the consumers. Finally, a limitation of the study is also the use of Greek participants only and not from other countries as well. Different cultures, especially outside the Mediterranean area, are expected to have minor differences on the traceability perception of TFs. This is the first study on understanding the traceability parameters of TFs for purchase and consumption in the new period after the pandemic crisis from the consumers' point of view.

Despite the importance of our findings, additional studies are needed in order to investigate further the parameters of traceability in the TFs, the long-lasting effects, and adaptations behavior to the "new normality". The findings contribute further to the main objective, which is the integration of TFs into the daily food consumption in the countries where there is the potential for increased production such as Greece. They also contribute to economic policies interventions required aimed at supporting increased production of TFs in Greece and elsewhere as they are important key factors for regional and territorial development, especially in inner and marginal areas. Further studies should expand in two different directions: studying TFs of other countries EU primarily, either themselves or in comparison, and studying the concept of traceability through the in-depth investigation of other pieces of information for Greek TFs perceived positively by the consumers.

**Author Contributions:** Conceptualization, supervision, and methodology, D.S.; writing—original draft preparation, D.S. and I.S.K.; investigation, E.C.; review and editing, D.S., T.B. and H.C.K. All authors have read and agreed to the published version of the manuscript.

**Funding:** This research received no external funding.

**Institutional Review Board Statement:** Not applicable.

**Informed Consent Statement:** Not applicable.

**Conflicts of Interest:** The authors declare no conflict of interest.

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
