# Peer review of "Consumers’ Perception on Traceability of Greek Traditional Foods in the Post-COVID-19 Era"

_sustainability, doi:10.3390/su132212687_

Round 1

Reviewer 1 Report

The research is interesting, well exposed and deals with a highly topical issue.

The parts of the paper are well articulated and developed according to a correct exhibition scheme. References are complete and relevant.

I would propose some simple observations in order to improve the proposed study:

  • I suggest to insert, in the first part of the study, some macroeconomic information about the economic weight (absolute and percentage values) of "Traditional Food" in the economy of agri-food products in Greece. Furthermore, the concept of "Traditional Food" could be extended as established by the European Union.
  • The limitations of the study should be better highlighted in the conclusions, among which the fact of having worked only on Greek TFs cannot be considered, given that it is the object of the study.
  • I would refer to possible economic policy interventions aimed at supporting such productions as they are important tools for territorial development, especially in inner and marginal areas.

Reviewer 2 Report

Considering the nascent field of traditional food traceability, the range of extant literature covered is apt and wide. I would welcome just a few more contemporary works, such as:

 “A values framework for measuring the influence of ethics and motivation regarding the performance of employees”. Business & Entrepreneurship Journal, Volume 10, Issue 1, pp 1-19

“Cash Holdings, Corporate Performance and Viability of Greek SME: Implications for Stakeholder Relationship Management”. EuroMed Journal of Business, Vol. 15 No. 3, pp. 333-348.,

Reviewer 3 Report

A good paper. The aim, hypothesis and consequently the findings can be described in a clearer manner.
